# Cracking Behavior of Gd₂Zr₂O₇/YSZ Multi-Layered Thermal Barrier Coatings Deposited by Suspension Plasma Spray

**Mohamed Amer** [1], **Nicholas Curry** [2], **Qamar Hayat** [1], **Rohit Sharma** [1], **Vit Janik** [1], **Xiang Zhang** [1], **Jon Nottingham** [3] and **Mingwen Bai** [1,*]

1    Centre for Manufacturing and Materials, Coventry University, Coventry CV1 5FB, UK
2    Thermal Spray Innovations, 5662 Salzburg, Austria
3    CN Technical Services Ltd., Wisbech PE13 2XQ, UK
*    Correspondence: mingwen.bai@coventry.ac.uk

**Abstract:** A new multi-layered thermal barrier coating system (TBCs) containing gadolinium zirconate (GZ, Gd₂Zr₂O₇) and yttria-stabilized zirconia (YSZ) was developed using suspension plasma spray (SPS) to improve the overall thermal cycling performance. This study focuses on the cracking behavior of the GZ/YSZ TBC after thermal exposure to find out the key factors that limit its lifetime. Different cracking behaviors were detected depending on the thermal treatment condition (i.e., horizontal cracks within the ceramic layer and at the thermally grown oxide (TGO)/YSZ interface) which can be related to stresses developed through thermal expansion mismatch and increased TGO thickness beyond a critical value, respectively. A reduction in hardness of bond coat (BC) was measured by nanoindentation and linked with the thermally activated grain growth mechanism. The hardness and elastic modulus of ceramic layers (GZ and YSZ) showed an increased trend after treatment that contributed to the interfacial cracks.

**Keywords:** gadolinium zirconate (GZ); thermal barrier coatings (TBCs); multi-layered structure; suspension plasma spray (SPS); microstructure; hardness; elastic modulus

## 1. Introduction

Thermal barrier coatings (TBC) are widely utilized to protect the hottest sections of gas turbine engines [1–3], internal combustion engines components [4,5], etc., resulting in efficiency improvement as well as economic and environmental benefits such as reduced fuel consumption and carbon dioxide emissions, respectively [2,6]. Yttria-stabilized zirconia (YSZ) is the most commonly used top coat in TBC [7]. However, increasing demand for higher efficiency leads to increased working temperature of gas turbines (>1200 °C); therefore, various issues associated with YSZ-TBC need to be addressed, for example, high sintering rate [8–10], loss of phase stability [11–14], hot corrosion, and calcia–magnesia–alumina–silicate (CMAS) attack [15–18]. Gadolinium zirconate (Gd₂Zr₂O₇ or GZ) is a promising alternative to YSZ as it is characterized by lower thermal conductivity and sintering rate, and improved phase stability and capabilities to withstand CMAS attack [19–22].

Despite the abovementioned advantageous features, GZ also has undesirable properties that negatively affect its thermal cycling (TC) performance. GZ is characterized by a low coefficient of thermal expansion (CTE) that causes increased residual stresses due to the mismatch with the CTE of the metallic layers (bond coat (BC) and substrate) [9]. Also, GZ has a high tendency to react with the alumina scale (thermally grown oxide, TGO) at higher temperatures to form GdAlO₃ [23], resulting in the consumption of the protective TGO layer. Furthermore, GZ has relatively lower fracture toughness as compared to YSZ [8,9]. For these issues, GZ exhibited shorter thermal cyclic lifetime as compared to YSZ [24]. Multilayered structure designs of TBC combining GZ and YSZ is a valid approach to remedy those weaknesses and enhance the performance of the overall TBC system [7,8,25–27].

For instance, Bakan et al. [24] reported an enhanced thermal cyclic lifetime of GZ/YSZ double layer TBC deposited by an atmospheric plasma spray (APS) process compared to single layer TBCs of GZ and YSZ. However, Doleker et al. [28] reported that double layered TBC (YSZ/Gd$_2$Zr$_2$O$_7$) deposited by an electron beam physical vapor deposition (EB-PVD) technique did not show good performance due to micro cracks after 50 h and 100 h oxidation tests at 1000 °C. Shen et al. [29] investigated the thermal shock resistance and failure behaviors of La$_2$Zr$_2$O$_7$/YSZ, La$_2$Ce$_2$O$_7$/YSZ, and Gd$_2$Zr$_2$O$_7$/YSZ deposited by EB-PVD and arc ion-plating physical vapor deposition (AIP-PVD). The longest thermal shock life was exhibited by GZ/YSZ. It was concluded that thickness and crack evolution affect the failure response. More studies demonstrated that multi-layered TBCs structure showed a longer lifetime compared to single-layered TBCs [30–32]. Further investigations studied the failure behavior of multi-layered TBCs corresponding to various factors such as TGO thickness, CMAS attack, oxidation, and thermal cyclic response [33–37]. Zhong et al. [38] studied the failure behavior of GZ/YSZ double structure TBCs deposited by APS technology and reported the spallation of the GZ layer at the GZ/YSZ interface owing to lower fracture toughness of GZ than YSZ. Yet, it seems that the development of the multilayered structure TBC has hit a bottleneck as there is a need for more in-depth investigations to understand the behavior of each individual layer and their synergy. Therefore, linking between microstructure changes and mechanical properties evolution of coatings' materials before and after applying different thermomechanical loading scenarios is an important step towards better understanding the coatings' behavior and the potential for the enhancement of its performance [39,40].

In this study, four TBC systems were deposited by suspension plasma spray (SPS) (Figure 1). (1) Double layer (DL) TBC comprising Gd$_2$Zr$_2$O$_7$ and YSZ layers. (2) Triple layer (TL) TBC consisting of a dense GZ top layer that can provide better resistance to CMAS attack and erosion, GZ intermediate layer, and YSZ base layer. (3) Composite layer (CL) TBC that include a dense GZ top layer, GZ + YSZ composite intermediate layer, and YSZ base layer. (4) Single layer (SL) TBC composed of YSZ as a benchmark. The performance and durability of these coatings were assessed with a focus on the fracture toughness, erosion performance, thermal conductivity, and thermal shock lifetime [7,8,27,41,42]. All the multi-layered TBCs had a longer thermal cyclic life compared to single layer (SL) TBC. The composite layer (CL) has relatively better fracture toughness and erosion performance as compared to triple layer (TL) TBC. This study is a continuous work with in-depth investigations on the failure analysis of these multilayered TBCs using a combination of SEM/EDX/EBSD/XRD and nanoindentation techniques to better understand the failure behavior of multilayered TBCs aiming for a better coating design.

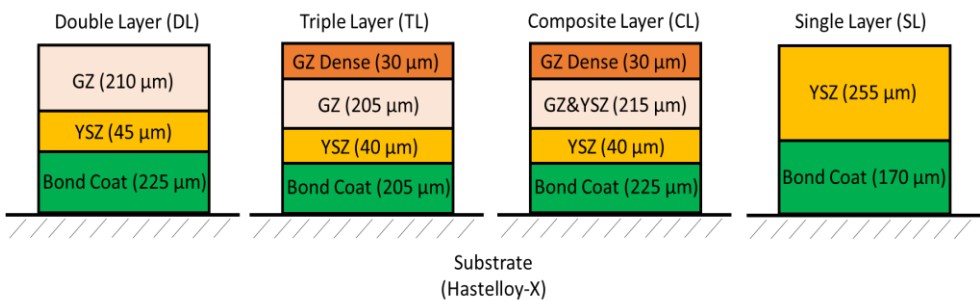

**Figure 1.** Schematic representation of the investigated multilayer and single layer TBC.

## 2. Experimental Methods

### 2.1. Samples Preparation

Bond coats were produced using the high velocity air fuel (HVAF) process (UniqueCoat M3, UniqueCoat LLC, Oilville, VA, USA) using a commercial bond coat material, AMDRY 386 (Oerlikon Metco, Wohlen, Switzerland). Substrates were Hastelloy-X plates that were cleaned and then grit blasted to achieve a surface roughness of ~3 ± 1 µm on Ra scale.

Before bond coat deposition, samples were air blasted to remove alumina grit particles sticking on the surface of substrates. A spray distance of 350 mm with nitrogen as the carrier gas was used to deposit the bond coat on the substrates. The bond-coated substrates were preheated at a temperature of 200–250 °C using a plasma gun operated without the suspension. Preheating could help in removing the volatile impurities from the bond-coated surface. An Axial III plasma system with a Nanofeed 350 liquid feeder (Northwest Mettech Corp., Surrey, BC, Canada) was used to produce the GZ/YSZ layers. The suspensions for this study were produced by Treibacher Industries AG, (Althofen, Austria) and consisted of an 8 wt.% YSZ powder (8YSZ) and a GZ powder with an average particle size of approximately 550 nm and dispersed as a suspension. The suspensions were formulated with a solids load of 25 wt.% in an ethanol-based solvent. The composite layer coating was produced using a mix of these two suspensions at a ratio of 50:50. The dense GZ outer layer was produced from a suspension formulated in a water-based solvent at 40% solids load. To deposit the columnar structure, a standoff distance of 100 mm was maintained, whereas for the deposition of the GZ dense layer, a lower standoff distance was kept (i.e., 70 mm). The spray power used to produce the columnar structure was 125 kW, whereas for the denser GZ top layer, 85 kW was utilized. It is worth mentioning that a GZ layer of approximately 11 μm thick was deposited in single spray pass. While in case of YSZ, a layer approximately 10 μm thick was sprayed in one pass. Further information for the preparation and testing of these coatings can be found in the literature [7–10].

### 2.2. Thermal Treatment

The TBC samples (dimension 20 mm × 10 mm × 2 mm) were exposed to 1150 °C for 1, 10 and 100 h at an air atmosphere condition inside a Lenton furnace (AWF 12/12, Hope Valley, UK). Note that the thermal exposure temperature is an accelerated test and is around 150 °C higher than the designed operating level of the bond coat. The average heating and cooling rates were 10 °C/min and 1.5 °C/min, respectively. The furnace was calibrated using a thermocouple before usage, and the error percent in the measured temperature inside the furnace was ±0.5.

### 2.3. Material Characterisation

#### 2.3.1. Metallography

The coatings were cross sectioned using a SiC cutting wheel on a Struers secotom-50 machine at a relative slow feed speed (0.07 mm/s) to avoid damage of coatings during the sectioning process. It is worth mentioning that the 10 h and 100 h samples were first epoxy molded to avoid introducing cracks during handling and metallography preparation procedures. Cross-sections were mirror polished down to 50 nm using a colloidal silica polishing suspension on Buehler AutoMet™ 300 (Lake Bluff, IL, USA). Subsequently, the mirror-finished TBC samples were carbon sputtered (Quorum Q150T ES, Lewes, UK) to reduce charging during SEM analysis.

#### 2.3.2. Microstructure

The TBC specimens were examined with a field emission FEG-SEM (Zeiss Sigma 500 VP, Oberkochen, Germany) using back scattered electron (BSE) mode at 20 kV. Energy dispersive spectroscopy (EDS) was used to study the TGO growth and its chemical composition.

EBSD analysis was performed on a Symmetry S2® EBSD detector (Oxford Instruments, Abingdon, UK) to observe the grain size evolution, grain orientation, and phase distribution at a step size of 0.03–0.15 μm, and minimum of 800 grains per map was measured.

#### 2.3.3. XRD Analysis

XRD analysis was carried out on the top surface using a D8 Advance DaVinci system (Bruker, Coventry, UK) and a LYNXEYE detector with a Cu-K$\alpha$ radiation of 1.54 Å wavelength from 20° to 90°, and a step size of 0.01° and 0.12 s as the time per step. Quan-

titative Rietveld refinement (TOPAS V5, Bruker, Mannheim, Germany) was employed to determine the crystallite size based on the principles of whole powder pattern modelling (WPPM) [43]. The crystallographic information files (CIF) for the YSZ (ICSD 01-082-1242) and GZ (ICSD 01-080-0471) were imported into the TOPAS software package as starting models for the Rietveld refinement. Rietveld analysis was performed by refining the crystal parameters and calculating the crystallite size to achieve a good of fitness (GOF) within 5% between the experimentally determined X-ray diffractions and the corresponding calculated diffractions. The crystallite size was calculated according to the Lorentzian peak fitting function available in the TOPAS software. Different factors affect peak broadening including instrumental and sample contributions (e.g., source of radiation, misalignment of the diffractometer, crystallite size, micro-strain, structural faults, etc.). However, in the conducted analysis, it is considered that the peak broadening comes from the crystallite size evolution. Since all the samples were tested using the same XRD machine and testing parameters, the instrumental effect contribution is equal for all the samples.

### 2.4. Mechanical Characterization

ZHN nanoindenter (ZwickRoell, Worcester, UK) was used to characterize the mechanical properties (i.e., hardness and elastic modulus) of the TBCs different segments before and after applying the thermal treatment. The mechanical properties were evaluated using contact mechanics equations and the Oliver-Pharr method [44]. Force-controlled indentation tests were performed using a Berkovich tip (E = 1141 GPa; $\nu$ = 0.07). The parameters used in the performed nanoindentation tests are maximum load = l00 mN, loading rate = 16.63 mN/s, unloading rate = −45 mN/s, and dwell period at maximum load = 40 s. Typically a distance of 50 μm between indentations was maintained to avoid any interference effect from plastic deformation zones around indents. At least 10 indentations and up to 30 indentations, depending on the thickness of each layer, were carried out for the investigated samples, and the average values were reported.

The obtained data were statistically analyzed by one-way analysis of variance (ANOVA) in OriginLab software. The performed ANOVA analysis helps to judge the obtained data if there is a significant difference or not. Statistical significance was reported at $p$, 0.05 (*), $p$, 0.01 (**), $p$, 0.001 (***), and $p$, 0.0001 (****), unless otherwise stated.

## 3. Results and Discussion

### 3.1. Microstructure

The cross-sectional SEM images of DL samples revealed a columnar microstructure (Figure 2a–d). The thickness measurements for the DL samples layers (GZ/YSZ/BC) were determined using Image J software [45]. The top layer (i.e., GZ) thickness was measured to be 210 ± 16.8 μm, whereas the YSZ was measured to be 43.2 ± 5.3 μm and the BC was 225.5 ± 11.2 μm. Figure 2c reveals that horizontal cracks developed within the GZ segment at nearly 60 μm away from TGO scale after 10 h. While Figure 2d illustrates cracking of DL-TBCs through the TGO/YSZ interface after 100 h. Furthermore, linking up of the cracks developed within the GZ layer, and the cracking that occurred through the YSZ/BC interface resulted in delamination of the coating as shown in Figure 2c.

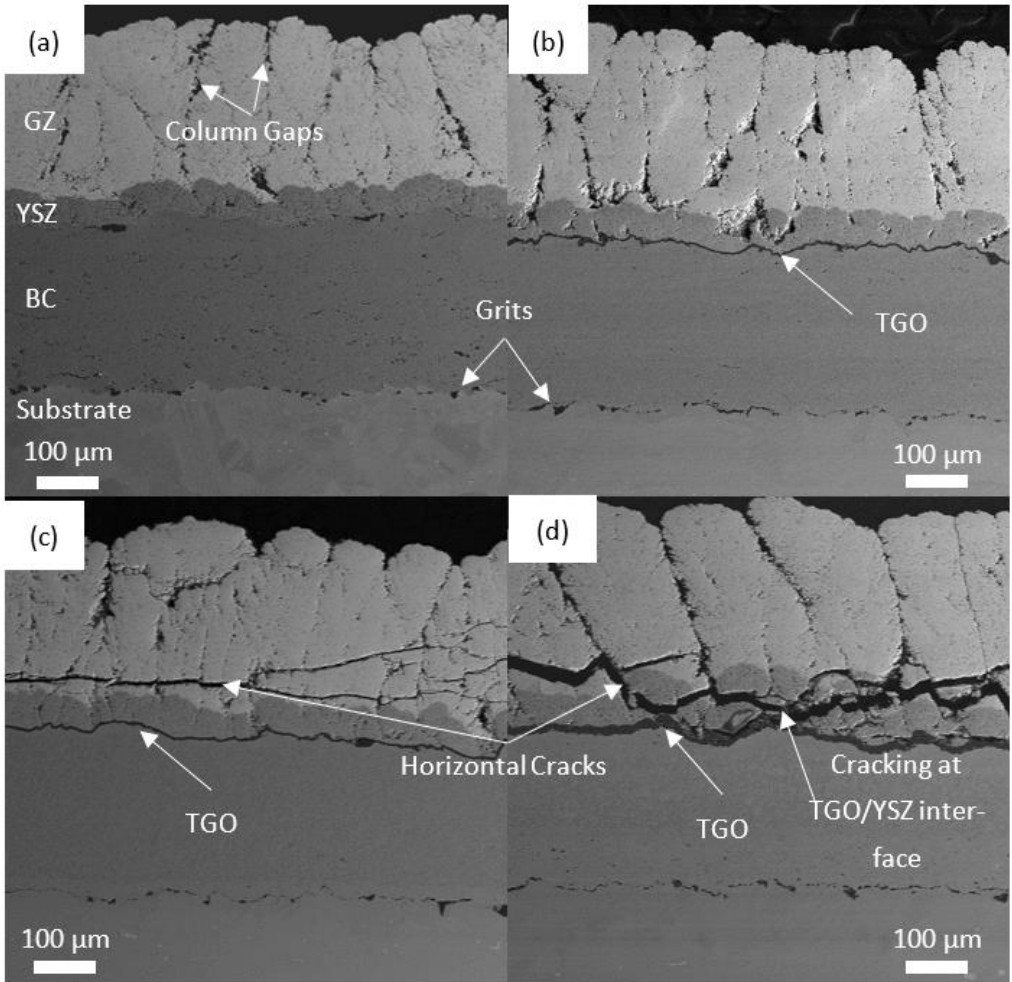

**Figure 2.** BSE of the double layered TBC (**a**) as sprayed, (**b–d**) 1, 10, and 100 h, respectively.

The TGO thickness for 1, 10, and 100 h were measured to be $3.2 \pm 0.4\ \mu m$, $3.4 \pm 0.5\ \mu m$, and $10 \pm 1.4\ \mu m$, respectively (Figure 3). Dong et al. [46] investigated the TGO thickness effect on the failure of plasma-sprayed TBCs and reported that the critical TGO thickness was nearly 6 μm. The cracking behavior (Figure 2c) is recognized at a TGO thickness below the critical value and can be attributed to thermally induced residual stresses developed within TBCs during thermal loading due to thermal expansion mismatch [46]. For instance, the thermal expansion coefficient (CTE) of ceramic layers (about $10.4 \times 10^{-6}\ K^{-1}$ and $11 \times 10^{-6}\ K^{-1}$ for GZ and YSZ, respectively [9]) is typically lower than that of both the substrate and bond coat (about $15 \times 10^{-6}\ K^{-1}$ [47]), which leads to higher thermal stresses in the TBC system [46,48]. The developed horizontal cracks can directly contribute to spallation and delamination of the coatings from the substrate surface. Beyond critical thickness of TGO, this layer results in strain incompatibilities and mismatch stresses, thereby the TBCs system is subjected to damage and final delamination. The alternate failure behavior (Figure 2d) is evident at a TGO thickness greater than the critical value. This failure can be related to the oxidation-induced volume increase of the TGO and the variances in the thermal expansion between TGO and the remaining TBC layers [46,48]. It is known that TGO growth is one of the most important factors that lead to the failure of TBCs due to the induced stress and strain fields. The growth of TGO layer considerably changes stress intensity and its distribution [46]. For instance, compressive stresses are present at the valleys and tensile stresses are present at the peaks with thin TGO, whereas the nature of the stresses in the ceramic layer (YSZ) valley alternates from compression to tension as TGO becomes thicker. Hence, TGO growth can significantly affect the interface

structure between the top and bond coat leading to reduced interfacial adhesion strength and delamination of the coatings [46]. Depending on temperature and time, the TGO scale grows as the aluminum in β-NiAl phase depletes, which leads to phase transformation within the bond coat from β to γ/γ′ phases. Also, as a result of β-NiAl phase depletion, metal oxides such as $Cr_2O_3$, NiO, CoO, or spinel $(Co, Ni)(Cr, Al)_2O_4$ can be formed [28]. The EDS elemental mapping for the TGO regions at different thermal exposure duration of DL-TBC samples are shown in Figure 3a–c. Based on the EDS analysis, alumina (i.e., $Al_2O_3$) is the dominant metal oxide in the TGO layer, whereas very little spinel or mixed oxide was present (overlapped Co, Cr, Ni, and O trace) just above the continuous alumina layer in the center (Figure 3b). The same observations were detected for the other TBC samples (i.e., triple, composite, and single layer structures) investigated in this research work. Therefore, EDS analysis and the related discussion are only presented in this section for the DL-TBC samples.

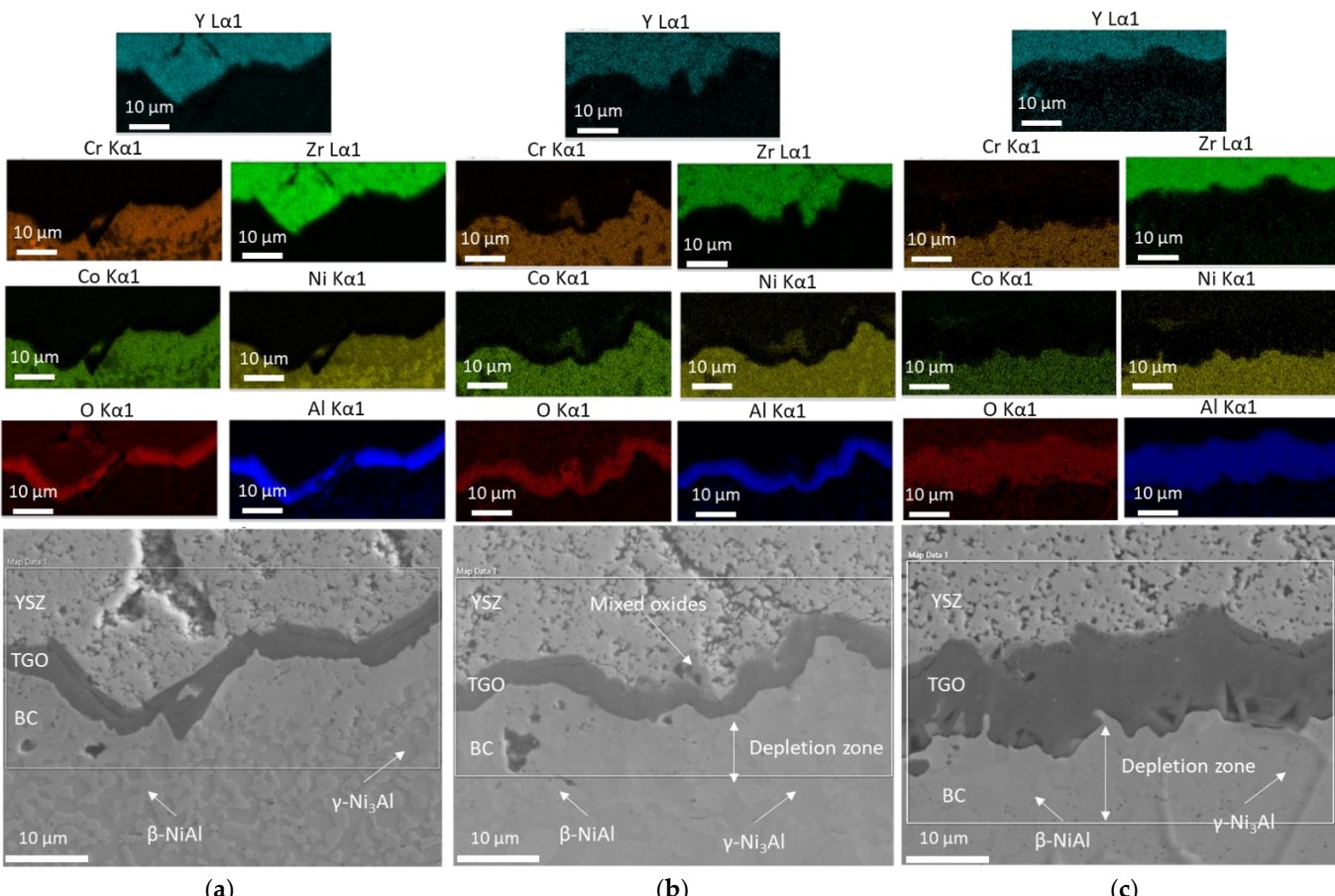

**Figure 3.** EDS elemental mapping of TGO region for the double layered DL TBC after (**a**) 1 h, (**b**) 10 h, (**c**) 100 h.

As seen in Figure 4a–d, the TL-TBC samples comprise a dense GZ top layer, which can be characterized by non-columnar, micro-gaps-free, and a relatively brighter contrast region. A columnar microstructure of the GZ and YSZ layers with in-between column gaps can be seen. The thickness of the GZ dense layer was measured to be 28.8 ± 2.8 μm. The intermediate GZ segment has a thickness of 203.4 ± 10.4 μm, whereas the thickness of YSZ (with dark contrast under BSE) is 41.8 ± 4.3 μm, and the bond coat is 205.2 ± 9.5 μm. The thickness of the TGO scale after 1, 10, and 100 h is 3.1 ± 0.5 μm, 3.4 ± 0.7 μm, and 10.5 ± 1.3 μm, respectively. Furthermore, horizontal cracking occurred for the TL-TBC system after being thermally loaded for 10 h (Figure 4c). These cracks were developed within the intermediate GZ layer and at a distance of nearly 130 μm away from the TGO

scale. Figure 4d shows the failure of TL-TBC after 100 h which can be recognized by cracking at the TGO/YSZ interface coupled with horizontal cracks developed within the GZ segment at approximately 60 µm away from the TGO scale. Delamination of the coating was observed after 100 h as the TGO thickness exceeded the critical value.

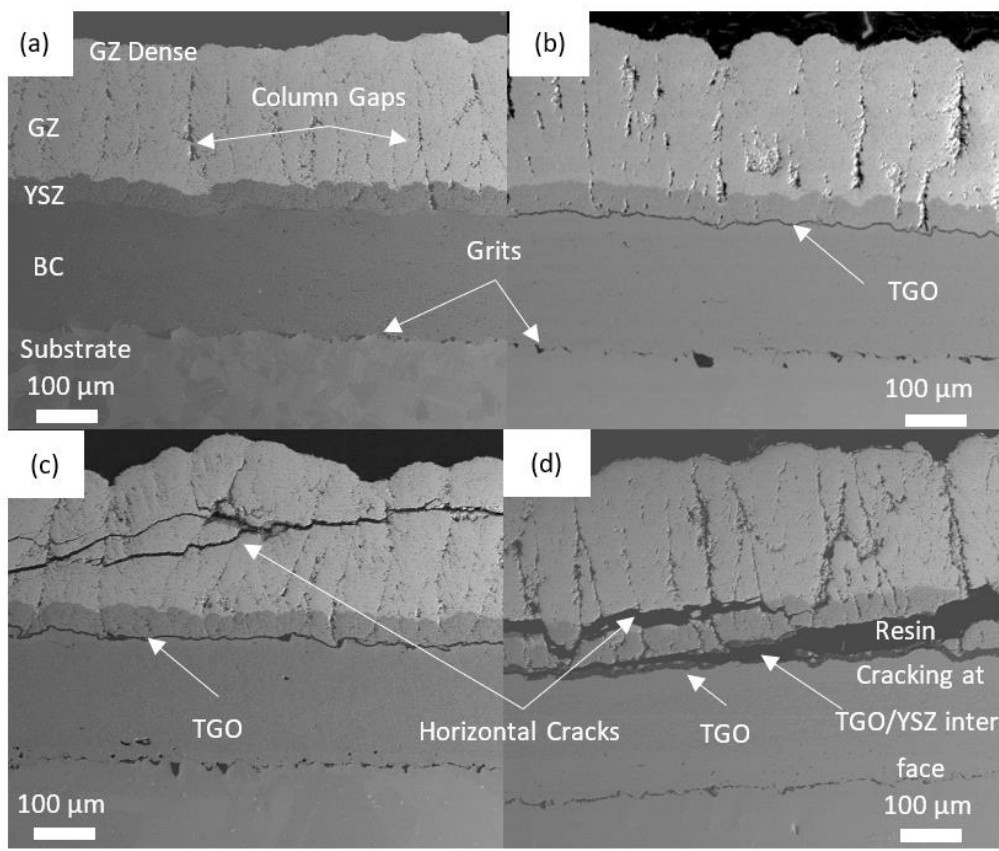

**Figure 4.** BSE of the triple layered TL TBC (**a**) as sprayed, (**b**–**d**) 1, 10, 100 h, respectively.

Figure 5a–d shows the composite TBC samples (GZ dense/GZ + YSZ/YSZ). The GZ dense top layer appears to have bright contrast under BSE with a thickness of 27.6 ± 4.2 µm. The intermediate composite layer (GZ + YSZ) was measured to be 214.1 ± 25.3 µm. Whereas the thickness of YSZ base layer and metallic bond coat are 41.5 ± 4.8 µm and 224.7 ± 9.6 µm, respectively. The thickness of TGO after 1, 10, and 100 h are 3.1 ± 0.3 µm, 3.6 ± 0.6 µm, and 9.7 ± 1.1 µm, respectively. The CL-TBC sample after 10 h experienced failure that can be seen as horizontal cracks generated within the intermediate GZ + YSZ layer (Figure 5c) 180 µm away from the TGO layer. After 100 h, the CL-TBC sample reveals cracking and delamination at the interface between YSZ and TGO layers.

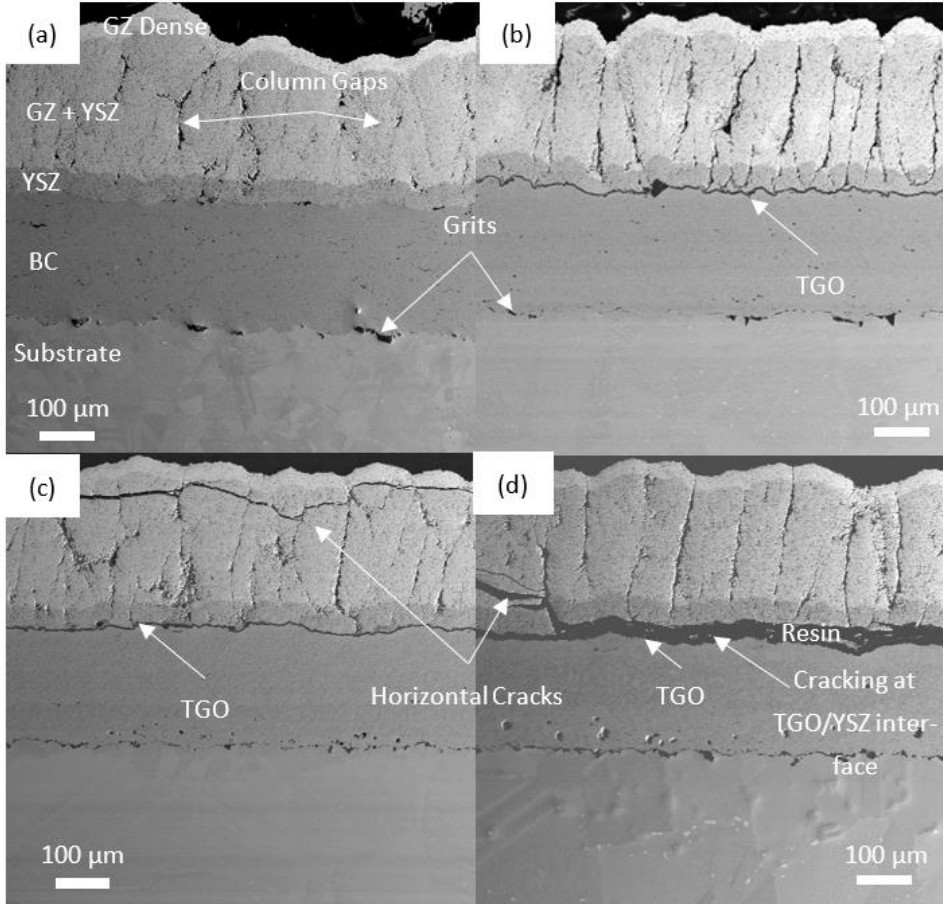

**Figure 5.** BSE of the composite layered TBC (**a**) as sprayed, (**b–d**) 1, 10, 100 h, respectively.

For comparison, Figure 6a–d shows the thickness of single layer YSZ is 256.1 ± 21.5 µm with a bond coat of 171.6 ± 8.8 µm, with a TGO thickness of 3 ± 0.5 µm, 3.4 ± 0.5 µm, and 11.4 ± 1.4 µm, respectively. After 10 h, a horizontal crack is observed in YSZ which is 60 µm from TGO (Figure 6c). Also, vertical cracks formed at the YSZ columnar interfaces can be observed, which can influence the thermal barrier efficiency. After 100 h, cracks are observed at the TGO/YSZ interface which leads to final delamination of the coatings (Figure 6d) with a TGO thickness above critical value.

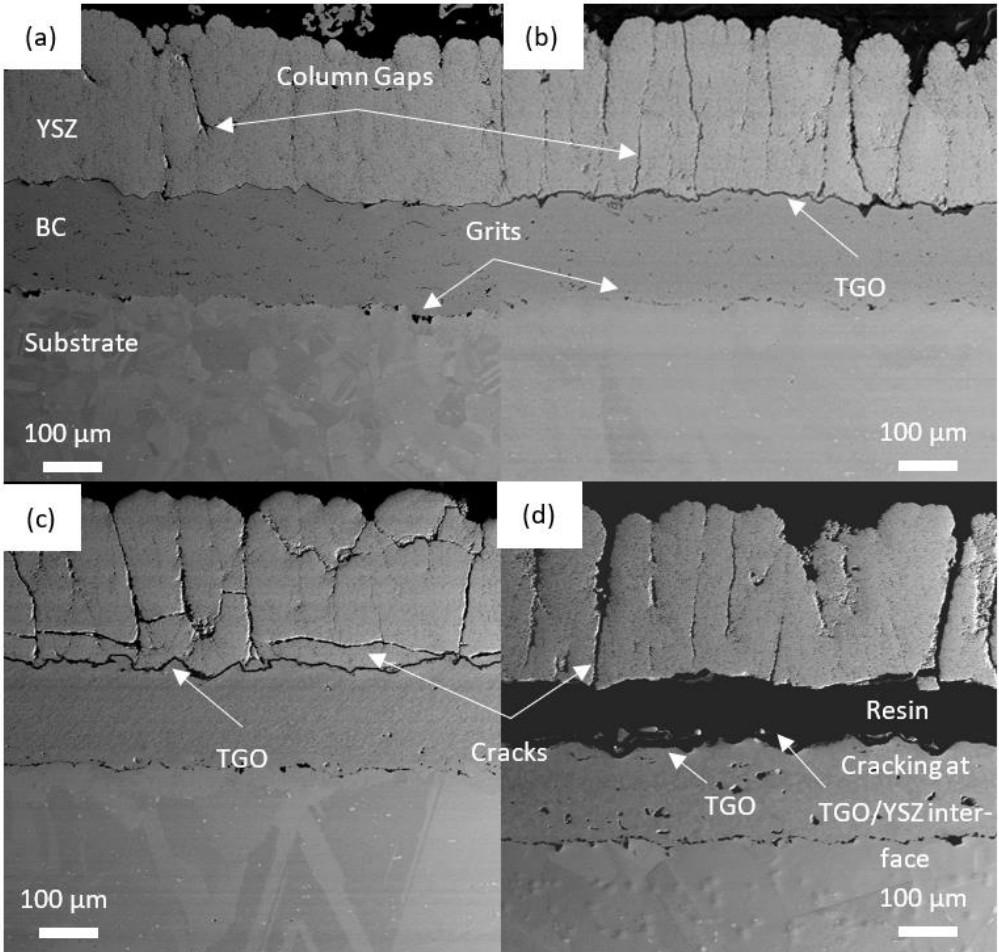

**Figure 6.** BSE of the single layered TBC (**a**) as sprayed, (**b–d**) 1, 10, 100 h, respectively.

*3.2. XRD Analysis*

Good stability of the cubic GZ phase can be noted alongside all the recorded XRD patterns (Figure 7a–c). For the SL-TBCs, t′-YSZ decomposed into tetragonal (t) and cubic (c) phases after exposure to the applied thermal loading duration (see Figure 7d), which can be recognized as one peak at 35 and 60 2 theta transformed to two peaks as shown in Figure 7d, and further explanation can be found in Bai et al. [49].

In addition, all peaks appear to be sharper after treatment indicting crystallite growth (Figure 8). For DL-TBC samples, crystallite growth in GZ layer is observed to be $60 \pm 1.3$ nm at the as-sprayed condition and increased to be almost doubled ($121.1 \pm 3.4$ nm) after 100 h. The same behavior was detected for TL-TBC samples as the crystallite size of GZ dense layer was measured to be $76 \pm 1.7$ nm for the as-sprayed condition and increased approximately two fold after heating for 100 h, $145.6 \pm 5.4$ nm. The crystallite size of CL-TBC specimens (GZ dense layer) for the as-sprayed and the 100 h treatment condition were determined to be $76.2 \pm 1.7$ nm and $152.5 \pm 5.4$ nm, respectively. For the SL-TBC samples, the crystallite size of YSZ layer (as-sprayed condition) was $61.8 \pm 0.8$ nm and increased to $122.6 \pm 2$ nm after 1 h, and no significant changes afterwards.

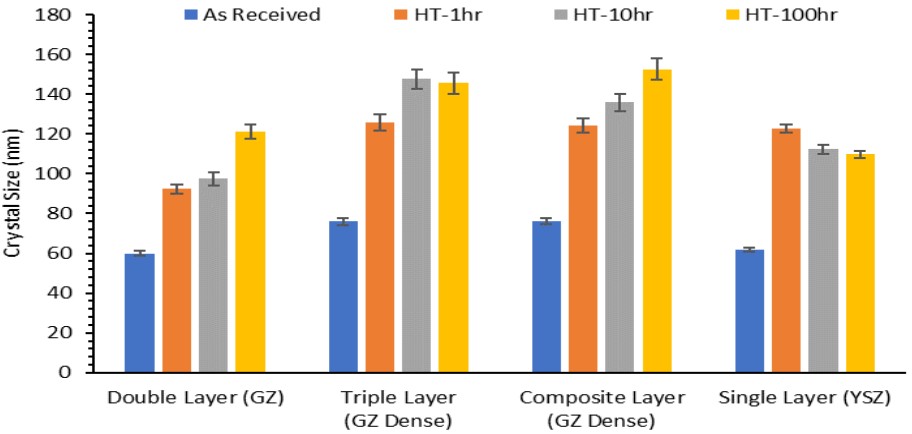

**Figure 7.** XRD of as-sprayed and heat-treated TBC samples (**a**) double layer, (**b**) triple layered, (**c**) composite layer, (**d**) single layer.

**Figure 8.** Crystallite sizes evolution of the investigated TBC systems (as-sprayed and heat-treated) obtained by Rietveld Refinement of the XRD patterns.

*3.3. EBSD Analysis*

EBSD was conducted to analyze grain size evolution, grain orientation, and phase analysis after thermal treatment. EBSD analysis was attempted on the as-sprayed samples; however, Kikuchi patterns for the layers of as-sprayed samples were too weak to be indexed (see Figure 9 as an example). This may be attributed to large residual stresses developed in the coating during the deposition process, its nano crystallinity structure, and some micro-strain retained within the coatings. Since HVAF was used to deposit the bond coat, the MCrAlY powder is deposited in a semi-molten state, with some of larger particles below their melting point. Deposition with high kinetic energy leads to a highly distorted structure. Therefore, it needs the initial thermal energy from the heat treatment to crystallize. Whereas for the as-received ceramic layers (i.e., YSZ and GZ), no clear EBSD mapping can be generated. This can be attributed to the complex stresses developed within the as-sprayed coating due to rapid cooling from plasma state temperature to room temperature, resulting in difficult EBSD mapping.

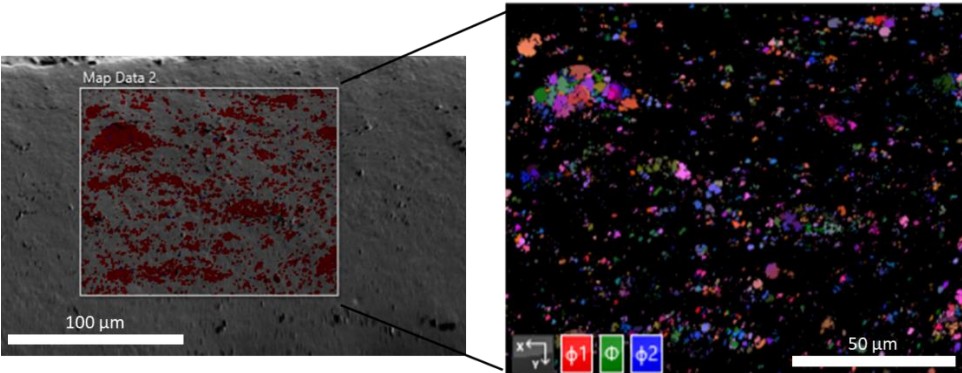

**Figure 9.** EBSD (Euler map) of bond coat for the as-received sample.

After heating, the TBC samples exhibited improved crystallinity, and EBSD mapping can be generated. The grain size distribution and orientation based on Euler angle maps for the DL-TBC after exposure for thermal loading of 1150 °C for 1, 10, and 100 h are shown in Figure 10. For the bond coat (Figure 10a), it was found that average grain size (dashed line) increased from $1.2 \pm 0.6$ µm to $1.6 \pm 0.8$ µm and $2 \pm 1.3$ µm, respectively. Also, the count of detected grains shown in the line graph (see Figure 10a), reduced from increased treatment duration from 1 h (4356 grain) to 100 h (1349 grain), is another indication for the induced grain-growth mechanism. We can conclude that application of thermal loading to the DL-TBC contributed to grain growth and was a prime mechanism aside from the grain formation mechanism. Furthermore, the EBSD maps obtained for the BC showed a balanced phase distribution of the β-phase and γ/γ'-phase; however, at the 100 h condition, a noticeable increase in the γ/γ'-phase percentage was detected. This phase transformation from β to γ-phase is accompanied with volume changes that can contribute to the failure of the TBC system [50]. For GZ/YSZ (Figure 10b,c), new grain formation is the dominant mechanism rather than grain growth as the average grain size (dashed line) remains at 0.3 µm and 0.5 µm, respectively. While the count of grains captured by EBSD scanning for YSZ at 1, 10, and 100 h increased from 1455 to 1740 and 1652 grains, respectively, the GZ grain count for 1, 10, and 100 h treatment duration were found to be 1105, 1344 and 1327, respectively. The total grain counts for ceramic layers were notably increased after 10 h and then stabilized after 100 h.

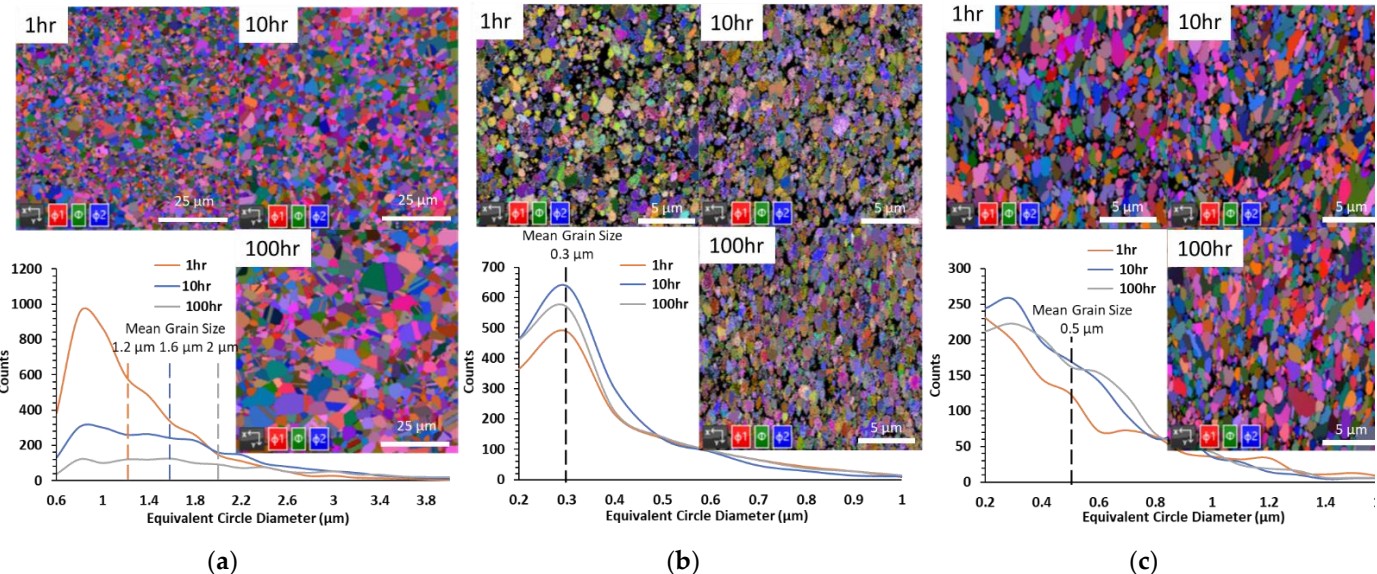

**Figure 10.** Grain size distribution and orientation for the double layer TBC (**a**) bond coat, (**b**) YSZ, (**c**) GZ layers.

For TL-TBC (Figure 11a–c), the grain size determined through EBSD analysis for the bond coat showed increased trends corresponding to the extended heating duration as shown in the line graph in Figure 11a. The average grain size measurement (represented by dashed lines in Figure 11a) obtained for the 1, 10, 100 h conditions are $1.2 \pm 0.6$ μm, $1.6 \pm 0.8$ μm, and $2.3 \pm 1.4$ μm, respectively. The metallic bond coat experienced grain coarsening that confirmed by the reduction in the count of indexed grains for the same recorded EBSD mapping area (grain count was 3947 and 1088 at 1 and 100 h, respectively), see Figure 11a. Mostly, the bond coat attained improved the crystalline structure after being thermally treated for 1 h, and further exposure for the thermal load contributed to the growth of the formed gains. The grain size distribution for the YSZ ceramic layer revealed insignificant changes between 1 and 10 h conditions; however, at the 100 h condition, a noticeable increase in the number of indexed grains were detected as presented in the line chart in Figure 11b. The average grain size for the YSZ layer was determined to be 0.3 μm. The grain size distribution (line graph, Figure 11c) for the GZ layer demonstrated an unnoticeable effect for the elongated exposure to heating load. However, it can be seen clearly from the Euler maps that grain growth is significant at the 100 h treatment condition. Also, the 10 h map shows black detected regions (unindexed grains), that may result from localized residual stresses. These unindexed regions might contribute to the unrealistic grain size distribution. The average grain size for the GZ layer were measured to be 0.5 μm, represented by the dashed line in Figure 11c. The orientation analysis from Euler maps, demonstrated that the YSZ and GZ grains have no preferred orientation.

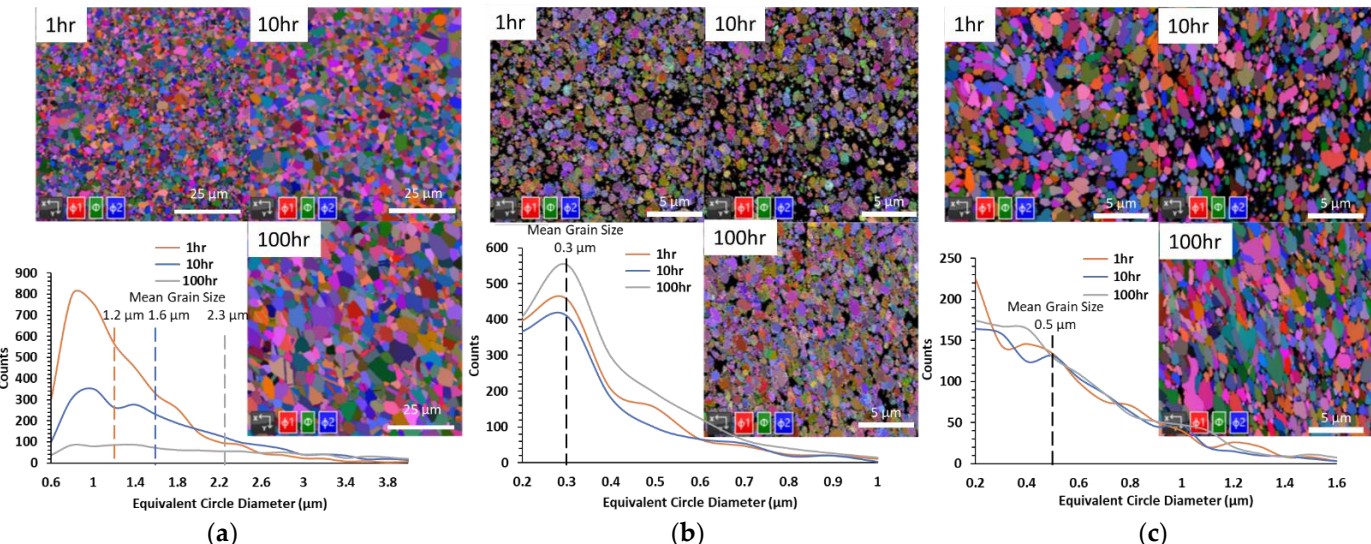

**Figure 11.** Grain size distribution and orientation for the triple layer TBC (**a**) bond coat, (**b**) YSZ, (**c**) GZ layers.

For CL-TBC (Figure 12), the average grain size (dashed lines) for the bond coat after heating for 1, 10, and 100 h was $1.1 \pm 0.6$ µm, $1.5 \pm 0.8$ µm, and $2.2 \pm 1.5$ µm, respectively. The number of grains recorded by EBSD Maps were reduced at 100 h condition (1119 grain) as compared to the 10 and 1 h treatment durations (2505 and 4392, respectively). The GZ + YSZ composite layer showed grain growth (Figure 12c) with the average grain size of $0.4 \pm 0.3$ µm, $0.5 \pm 0.4$ µm, and $0.6 \pm 0.4$ µm, respectively. Also, the grain count drops from 1705 at 1 h to below 800 grains at 10 and 100 h as another indication for the induced grain growth. While there is no significant change in YSZ layer (Figure 12b) as the average grain size of YSZ remains as 0.3 µm. The orientation analysis from Euler maps demonstrated that the YSZ and GZ + YSZ grains have no preferred orientation.

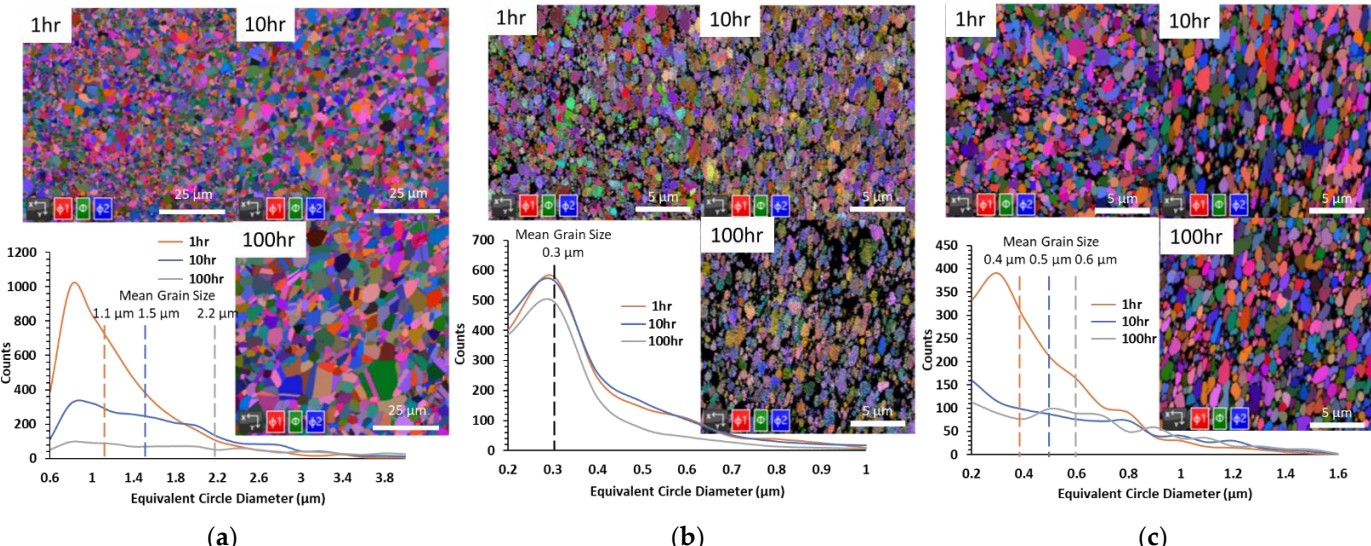

**Figure 12.** Grain size distribution and orientation for the composite layer TBC (**a**) bond coat, (**b**) YSZ, (**c**) GZ + YSZ layers.

For the SL TBC (Figure 13), the average grain size of BC is $1.2 \pm 0.7$ µm, $1.4 \pm 0.8$ µm, $2 \pm 1.5$ µm, respectively, with no significant changes in YSZ (Figure 13b) with the average grain size being 0.3 µm. The Euler maps show no preferred orientation for the YSZ grains.

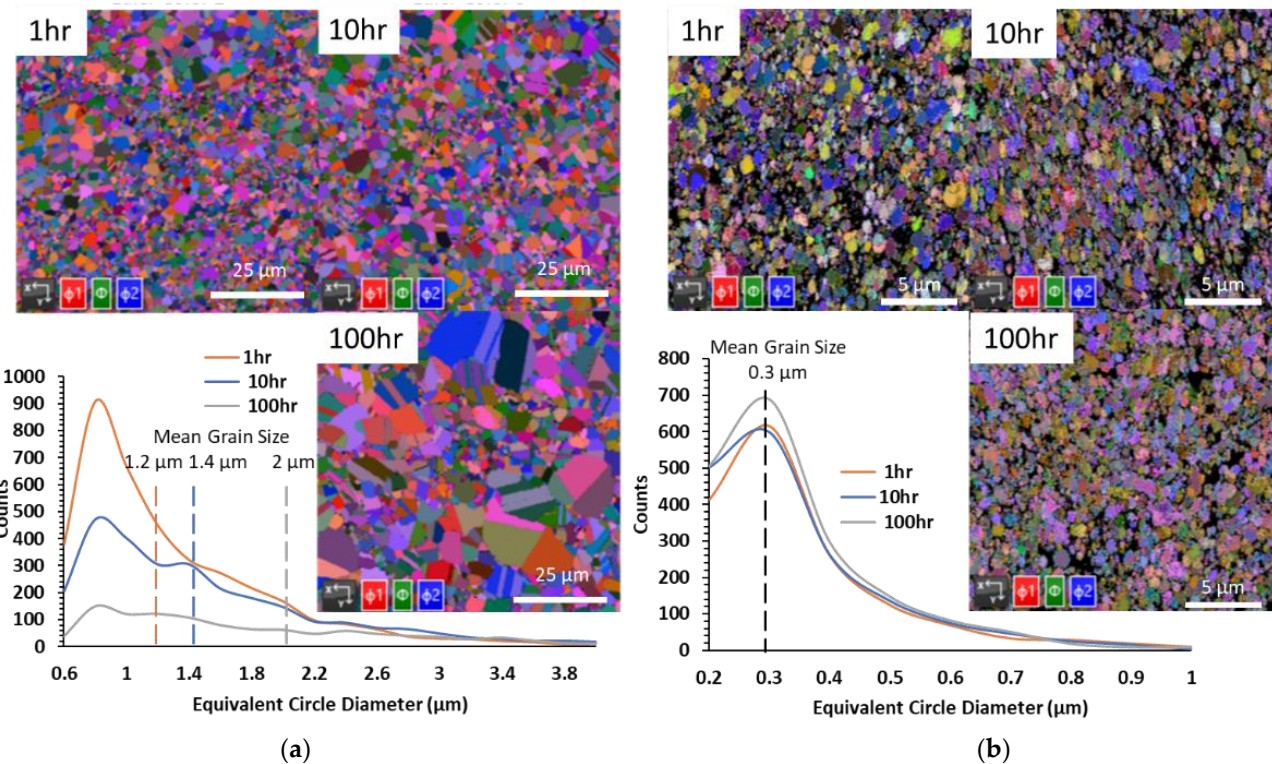

**Figure 13.** Grain size distribution and orientation for the single layer TBC (**a**) bond coat, (**b**) YSZ layers.

### *3.4. Evaluation of Mechanical Properties*

The hardness and elastic modulus measurements obtained for the DL-TBC samples are shown in Figure 14. The average hardness of the bond coat was significantly reduced from 8.69 ± 1.31 GPa at the as-sprayed condition down to 5.88 ± 0.31 GPa after 1 h and remained stable afterwards (Figure 14a). The high hardness is caused by its heavily strained structure along with the high residual stress developed during the deposition process. The elastic modulus measurement of BC layer experienced fluctuating response to the heating conditions as shown in Figure 14b. The elastic modulus of BC varied between 165.8 GPa (as sprayed) to 207.4 GPa (100 h treatment). This variation can be correlated to the microstructure changes observed before, that is diffusion of aluminum out of BC resulted in the transformation from β-phase to γ phase, which is also reported by Vignesh et al. [51]. The hardness measurements for the ceramic layers (YSZ and GZ) revealed an increased trend at 1 and 10 h, whereas there was no significant change after 100 h (Figure 14a). The hardness of YSZ increased from 6.44 GPa to 10.47 GPa, whereas the GZ increased from 6.22 GPa to 11.14 GPa at as-sprayed and 10 h, respectively. The same tendency was observed for the elastic modulus measurements of the DL-TBC samples as presented in Figure 14b. For the YSZ layer, the elastic modulus varied between 104.7 and 167.1 GPa, as-sprayed, and 10 h conditions, respectively. While the modulus of GZ layer changed from 106.3 GPa (as sprayed) to 176.2 GPa (10 h condition) which agree well with Zou et al. [52], who studied the performance of the suspension plasma-sprayed YSZ-TBC with different bond coat systems (i.e., APS and HVOF) after isothermal treatment at 1150 °C.

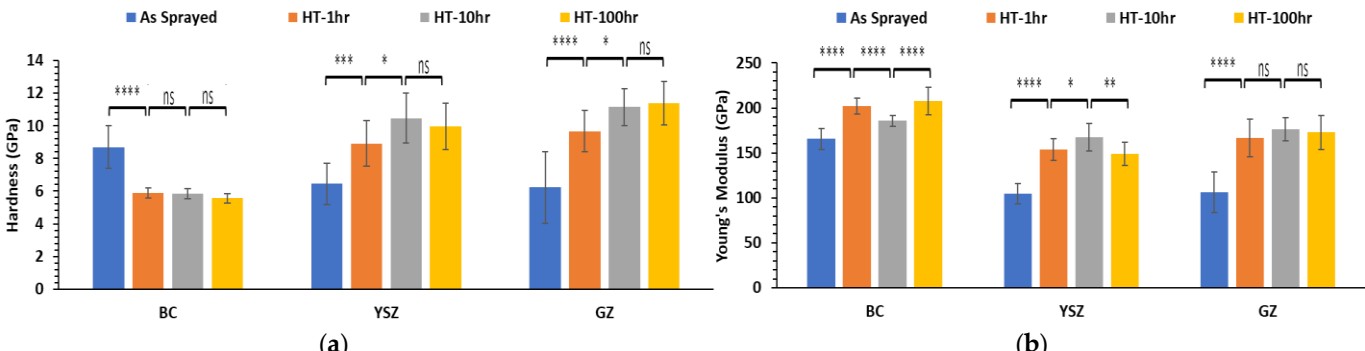

**Figure 14.** Nanoindentation data of the double layer TBC before and after thermal treatment (**a**) hardness (H), (**b**) elastic modulus (E). Asterisks indicate a statistical difference (* $p < 0.05$, ** $p < 0.01$, *** $p < 0.001$, **** $p < 0.0001$, as obtained using Tukey test, which computes a confidence interval for the difference between two means).

For TL TBC (Figure 15), similar behavior is observed in the bond coat as the previous one. Generally, there was an increase in hardness measurement for the ceramic layers (YSZ, GZ, and GZ density) as compared to the as-sprayed state as shown in Figure 15a. For example, the hardness of the YSZ layer increased from 5.01 GPa (as sprayed) to 8.87 GPa (10 h treatment), whereas there was an unnoticeable change in hardness for the 100 h treatment sample referred to in the 10 h specimen. For the GZ layer, the hardness was raised to 8.7 GPa after the 1 h heating condition (as-sprayed hardness = 5.31 GPa), whereas elongated exposure to the heating temperature (i.e., 10 and 100 h) had an unremarkable effect on determined hardness values. The hardness measurement for the GZ density fluctuated between 5.48 GPa and 11.02 GPa. The Young's modulus of the ceramic layers (YSZ, GZ, and GZ dense) increased after thermal treatment compared to the as-sprayed state (see Figure 15b). The elastic modulus for YSZ and GZ layer went up from 92.9 GPa to 122.8 GPa and from 101.6 GPa to 144.5 GPa at as-sprayed and 1 h treatment conditions, respectively. Whereas no significant change was detected for 10 and 100 h on elastic modulus of YSZ and. The Young's modulus values of the GZ dense layer varied between 93.9 GPa and 160.3 GPa.

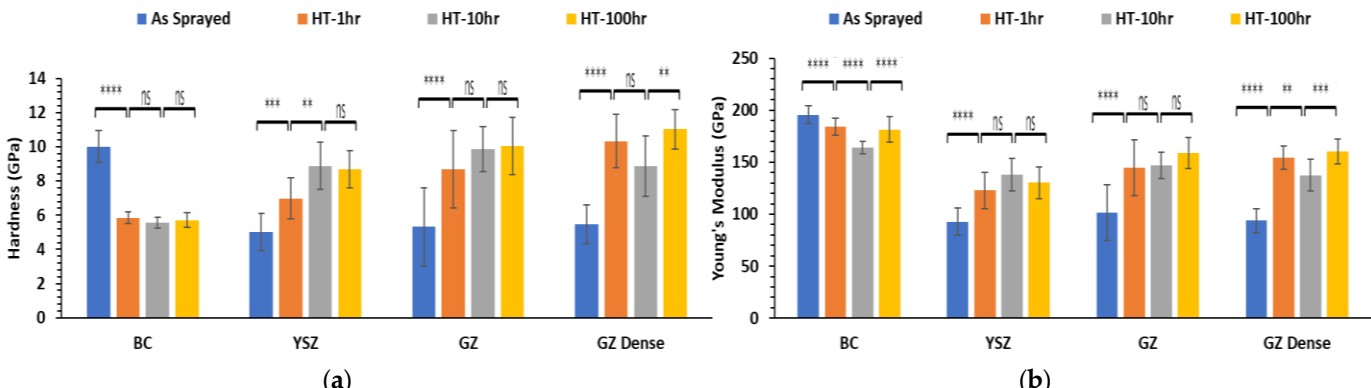

**Figure 15.** Nanoindentation data of the triple layer TBC before and after thermal treatment (**a**) hardness (H), (**b**) elastic modulus (E). Asterisks indicate a statistical difference (* $p < 0.05$, ** $p < 0.01$, *** $p < 0.001$, **** $p < 0.0001$, obtained using Tukey test, which computes a confidence interval for the difference between two means).

For CL TBC (Figure 16), there was no significant change in the hardness measurement of the YSZ layer through the comparison of each two successive conditions (e.g., as sprayed and 1 h, 10 h, and 100 h conditions). This is attributed to the variations in the hardness measurement of this layer (represented by the error bars). However, there is an increase in



the hardness of the YSZ layer after applying thermal treatment. For instance, the as-sprayed YSZ hardness was measured at 5.33 GPa and increased to 8.62 GPa at 100 h. The hardness of the composite layer (GZ + YSZ) was increased from 4.49 GPa (as sprayed) to 9.15 GPa (10 h), without noticeable difference in the hardness measurement between 10 h and 100 h. The GZ dense layer experienced an increase in the hardness from 5.16 GPa (as sprayed) to 8.49 GPa (1 h), without further significance in the obtained data for the remaining treatment conditions. The elastic modulus measurements for the ceramic layers were increased after thermal treatment as compared to the as sprayed state, see Figure 16b. For instance, the elastic modulus for the YSZ layer varied between 96.5 GPa (as sprayed) and 142.9 GPa (10 h condition). While the elastic modulus for the composite layer was measured as 82.7 GPa at the as-sprayed state and increased to 158.7 GPa after thermal treatment for 10 h. The Young's modulus of the GZ dense layer was altered within the range of 90.6 GPa (as sprayed) to 160.6 Gpa (10 h treatment).

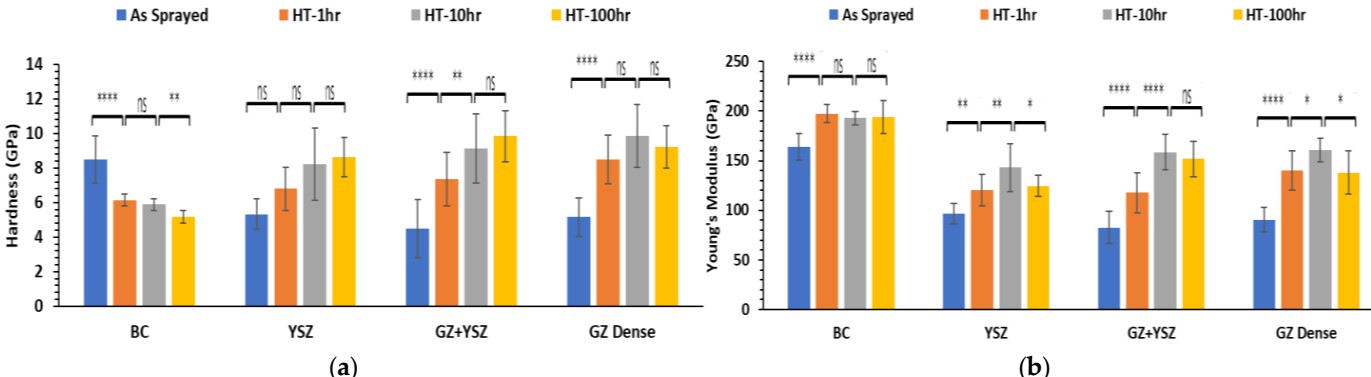

**Figure 16.** Nanoindentation data of the composite layer TBC before and after thermal treatment (**a**) hardness (H), (**b**) elastic modulus (E). Asterisks indicate a statistical difference (* $p < 0.05$, ** $p < 0.01$, *** $p < 0.001$, **** $p < 0.0001$, obtained using Tukey test, which computes a confidence interval for the difference between two means).

For SL-TBC (Figure 17), the observation of the changes occurred in hardness and elastic modulus measurements for both the metallic BC layer and the YSZ ceramic layer matches with the other coating systems discussed before. The detected increase in the mechanical characteristics of the ceramic layers can be attributed to the occurred healing of micro-cracks through induced sintering at high temperature along with the developed residual stresses within the ceramic layer due to thermal expansion mismatch [52,53].

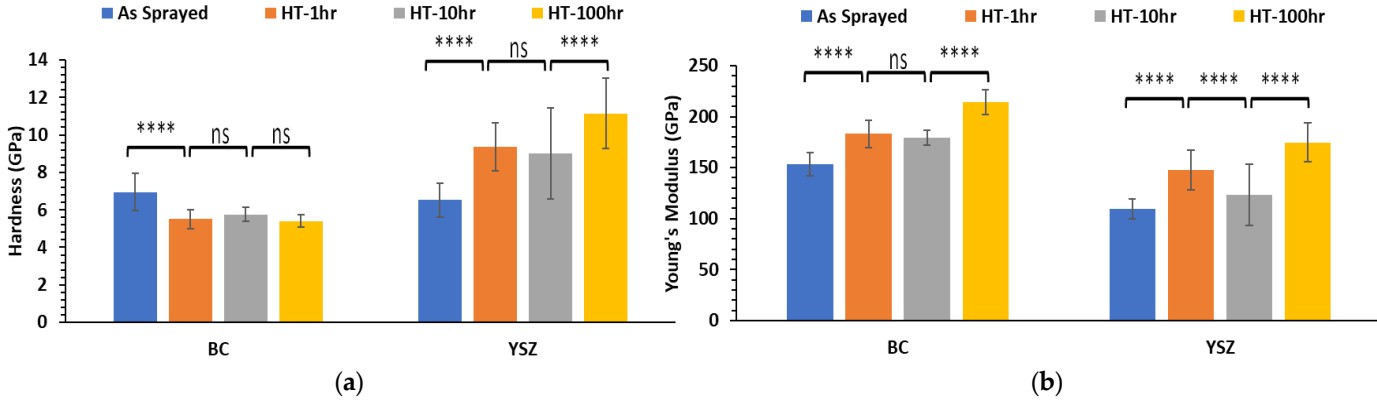

**Figure 17.** Nanoindentation data of the single layer TBC before and after thermal treatment (**a**) hardness (H), (**b**) elastic modulus (E). Asterisks indicate a statistical difference (* $p < 0.05$, ** $p < 0.01$, *** $p < 0.001$, **** $p < 0.0001$, obtained using Tukey test, which computes a confidence interval for the difference between two means).

## 4. Conclusions

In this work, four different layered TBC structures (i.e., double, triple, composite, and single layered) processed by SPS were subjected to a high temperature of 1150 °C for different durations (i.e., 1, 10, and 100 h). The changes in microstructure and mechanical properties of these coatings were studied using various characterization techniques (SEM, EDS, EBSD, XRD, and nanoindentation). The following conclusions can be drawn.

- Different cracking behaviors were detected for the 10 h and 100 h treatment conditions (horizontal cracks within ceramic layer and at TGO/YSZ interface, respectively). These cracking types can be attributed to thermal residual stresses developed within coatings due to thermal expansion mismatch and TGO thickness increase beyond critical value, accordingly.
- EBSD mappings revealed that grain growth and formation mechanisms were activated for TBCs layers, and phase transformation (β to γ phase) took place in the BC layer.
- The reduction in hardness of the BC layer was related to the observed grain coarsening mechanism, whereas elastic modulus variation is attributed to induced phase transformation. The increase in hardness and elastic modulus measurements for the TBC ceramic layers after thermal treatment, can be attributed to the sintering at high temperature as well as the developed residual stresses within the ceramic layers (GZ + YSZ) due to the mismatch in coefficient of thermal expansion (CTE) between the TBC layers.

**Author Contributions:** M.B. proposed this article idea and structured the paper in addition to revising the paper; N.C., produced the samples used in this study; M.A., Q.H. and R.S. were responsible for developing the work plan, data collecting, and analysis, as well as writing the paper; N.C., V.J., X.Z. and J.N. were responsible for revising and refining this article. All authors have read and agreed to the published version of the manuscript.

**Funding:** This research received no external funding.

**Institutional Review Board Statement:** Not applicable.

**Informed Consent Statement:** Not applicable.

**Data Availability Statement:** Data are contained within the article.

**Acknowledgments:** The authors would like to thank Zhaohe Gao, University of Birmingham for the help with sputter coating of the samples investigated in this research work.

**Conflicts of Interest:** The authors declare that there is no conflict of interest regarding the publication of this manuscript.

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
