# Peer review of "Cracking Behavior of Gd2Zr2O7/YSZ Multi-Layered Thermal Barrier Coatings Deposited by Suspension Plasma Spray"

_coatings, doi:10.3390/coatings13010107_

Round 1

Reviewer 1 Report

Review report: Cracking Behavior of Gd2Zr2O7/YSZ Multi-layered Thermal Barrier Coatings Deposited by Suspension Plasma Spray

1.       Shorten the length of the abstract section and add only key information in abstract section.

2.       Discuss the Novelty and clear application of the work in abstract as well as in introduction section.

3.       Shorten the length of the introduction section and add key published work and try to make a bridge between current and previous published work: https://doi.org/10.1016/j.ceramint.2018.01.131; https://doi.org/10.1007/s41779-018-0258-4; DOI 10.1088/2053-1591/ab5265.

4.       Provide the image of the experimental setup and clear information about coated sample and parameters.

5.       In xrd analysis, provide the quantitative analysis.

6.       How was the crystal size measured?

7.       Manuscript looks good but lacks in terms of technical discussion. Provide technical discussion with proper references.

8.       Discuss the crack formation mechanism near the coated surface.

9.       Shorten the length of the conclusion section. 

Author Response

Dear Reviewer,

The authors greatly appreciate the effort and time that the reviewer has devoted to providing us with such valuable comments and suggestions on our manuscript. All changes made in the manuscript are highlighted in red. Below are specific replies to reviewer #1 comments.

  1. Shorten the length of the abstract section and add only key information in abstract section.

Thanks for your suggestion, the abstract was shortened, and key information were added in the revised manuscript.

  1. Discuss the Novelty and clear application of the work in abstract as well as in introduction section.

Thanks for your valuable comment, this has been clarified in the abstract and introduction section of the revised manuscript.

  1. Shorten the length of the introduction section and add key published work and try to make a bridge between current and previous published work: https://doi.org/10.1016/j.ceramint.2018.01.131; https://doi.org/10.1007/s41779-018-0258-4; DOI 10.1088/2053-1591/ab5265.

Many thanks for your valuable comment and these valuable references, these references have been added to the revised manuscript (See reference 39-40).

  1. Provide the image of the experimental setup and clear information about coated sample and parameters.

More clarification regarding the coated samples experimental procedures and parameters were added to manuscript (sample preparation section).

  1. In xrd analysis, provide the quantitative analysis.

Thanks for your comment more clarification regarding the conducted quantitative analysis was added to the revised manuscript (XRD analysis section).

The crystallographic information files (CIF) for the YSZ (ICSD 01-082-1242) and GZ (ICSD 01-080-0471) were imported to the TOPAS software package as starting models for the Rietveld refinement. Rietveld analysis was performed by refining the crystal parameters and calculating the crystallite size to achieve good of fitness (GOF) within 5 % between the experimentally determined X-ray diffractions and the corresponding calculated ones. The crystallite size was calculated according to Lorentzian peak fitting function available in the TOPAS software. Different factors affect peak broadening including instrumental and sample contributions (e.g., source of radiation, misalignment of the diffractometer, crystallite size, micro-strain, structural faults, etc.). However, in the conducted analysis it is considered that the peak broadening coming from crystallite size evolution. Since all the samples were tested using the same XRD machine and testing parameters, the instrumental effect contribution is equal for all the samples.

  1. How was the crystal size measured?

The crystallite size measurements conducted in this research work was determined using Rietveld analysis technique through Lorentzian peak fitting function available in the TOPAS software.

More clarification was added to the revised manuscript in the XRD analysis section.

  1. Manuscript looks good but lacks in terms of technical discussion. Provide technical discussion with proper references.

Many thanks for your comment, the authors discussed all the findings thoroughly with citing supporting literature when appropriate. More discussion supported by publications were add in the revised manuscript when relevant.

  1. Discuss the crack formation mechanism near the coated surface.

Many thanks for your valuable comment, more discussion about the failure of the coatings were added to the revised manuscript (Results and discussion – Microstructure section).

Mainly, there are two cracking types were detected in the investigated TBC systems as a response for the exposure for thermal load of 1150 °C for different durations. The first cracking behavior was detected after exposure for 10 hr thermal treatment duration (i.e., horizontal cracks within the ceramic layers), whereas cracks at TGO/YSZ interface were developed after 100 hr treatment condition. These horizontal cracks can directly contribute to spallation and delamination of coatings out from substrate surface. Also, worthily mentioned that some vertical cracks were detected at the YSZ columnar interfaces for the 10 hr and 100 hr treatment conditions for the single layer YSZ TBC, which can influence the thermal barrier efficiency (see Figure 6 (c, d)). Dong et al. [1] investigated the TGO thickness effect on failure of plasma-sprayed TBCs and reported that the critical TGO thickness was nearly 6 µm. Hence, the first cracking behavior (10 hr) was detected at TGO thickness lower than critical value (report, ed in literature), and the crack formation can be attributed to the thermally induced residual stresses developed within TBCs during thermal loading due to thermal expansion mismatch between the TBC’ layers. For instance, the thermal expansion coefficient (CTE) of ceramic layers (about 10.4 × 10−6 K−1 and 11 × 10−6 K−1 for GZ and YSZ, respectively [2]) is typically lower than that of both substrate and bond coat (about 15 × 10-6 K−1 [3]), which leads to higher thermal stresses in the TBC system [1,4]. The second cracking behavior (cracking at TGO/YSZ interface) was detected at 100 hr thermal loading condition was driven by oxidation-induced volume increase of the TGO above critical value (approximately, 6 µm)  along with the differences in the coefficient of thermal expansion between TGO and TBC layers [1,4]. Also, it was noticed that microcracks linking up takes place at the 100 hr thermal treatment condition, which leads to final delamination of the coatings. It is known that TGO growth is one of the most important factors that lead to the failure of TBCs due to the induced stress and strain fields. The growth of TGO layer considerably changes stress intensity and its distribution [1]. For instance, compressive stresses are present at the valleys and tensile stresses at the peaks with thin TGO, while the stresses nature in the ceramic layer (YSZ) valley alters from compression to tension as TGO becomes thicker. Hence, TGO growth can significantly affect the interface structure between top and bond coat leading to reduced interfacial adhesion strength and delamination of the coatings [1].

  1. Shorten the length of the conclusion section.

Many thanks for your comment this have been modified in the revised version.

References

  1. Dong, H.; Yang, G.; Li, C.; Luo, X.; Li, C. Effect of TGO Thickness on Thermal Cyclic Lifetime and Failure Mode of Plasma‐sprayed TBC S. J. Am. Ceram. Soc. 2014, 97, 1226–1232.
  2. Gok, M.G.; Goller, G. State of the Art of Gadolinium Zirconate Based Thermal Barrier Coatings: Design, Processing and Characterization. Methods Film Synth. Coat. Proced. 2019.
  3. Vaßen, R.; Traeger, F.; Stöver, D. New Thermal Barrier Coatings Based on Pyrochlore/YSZ Double‐layer Systems. Int. J. Appl. Ceram. Technol. 2004, 1, 351–361.
  4. Vencl, A.A.; Mrdak, M.R. Thermal Cycling Behaviour of Plasma Sprayed NiCr-Al-Co-Y2O3 Bond Coat in Thermal Barrier Coating System. Therm. Sci. 2019, 23, 1789–1800.

Reviewer 2 Report

The paper is presenting the cracking behavior of GZ/YSZ multi-layered thermal barrier coatings, deposited by suspension plasma spray. It is an interesting subject that affects the attempts and the technology of thermal barrier coatings. The paper is well written with a comprehensive structure, however, there are some points that authors should look at:

1. For the cracking behavior: There are two types of cracks, developed at elevated temperatures: the vertical (normal to the layers) and the horizontal (parallel to the layer-interfaces). The vertical cracks may affect the thermal barrier efficiency; however, the horizontal ones may affect the coating detachment. Authors should make a clear remark on that, indicating the mechanism of crack formation. Authors state that the formation of cracks may be due to the difference of the thermal expansion coefficient of the different deposited layers, however, they are not supporting this claim with evidence (ex. what is the thermal expansion coefficient of each layer, if there is strain introduced at the interfaces due to lattice mismatch etc.)

2. For the crystalline size: Authors do not indicate whether they have evaluated the crystalline size using (for example) Scherrer equation. I assume that instrumental broadening has been taken into account. Scherrer equation assumes that peak broadening is only due to size. However, another possible mechanism is due to incoherent strain. Authors, should make a clear point on that.

3. For the grain size: Authors provide a mean value for the grain size. However, mean value implies that the distribution of grains is normal. Apart for the case of YSZ, where the distribution may be considered as symmetrical single-mode to some extent, in all other cases, it seems that grain distribution is multi-modal. A median value is more appropriate in these cases. It would be interesting if authors could make a peak analysis, and see whether some distributions are preserved despite the thermal treatment.

4. Minor corrections:

a. Authors do not state the composition of the GZ/YSZ composite.

b. In page 3, section “thermal treatment” authors do not indicate whether the treatment was performed in air or in inert or oxidizing atmosphere.     

Author Response

Dear Reviewer,

The authors greatly appreciate the effort and time that the reviewer has devoted to providing us with such valuable comments and suggestions on our manuscript. All changes made in the manuscript are highlighted in red. Below are specific replies to reviewer #2 comments.

  1. For the cracking behavior: There are two types of cracks, developed at elevated temperatures: the vertical (normal to the layers) and the horizontal (parallel to the layer-interfaces). The vertical cracks may affect the thermal barrier efficiency; however, the horizontal ones may affect the coating detachment. Authors should make a clear remark on that, indicating the mechanism of crack formation. Authors state that the formation of cracks may be due to the difference of the thermal expansion coefficient of the different deposited layers, however, they are not supporting this claim with evidence (ex. what is the thermal expansion coefficient of each layer, if there is strain introduced at the interfaces due to lattice mismatch etc.)

Many thanks for your valuable comment, more discussion about the failure of the coatings were added to the revised manuscript (Results and discussion – Microstructure section). 

Mainly, there are two cracking types were detected in the investigated TBC systems as a response for the exposure for thermal load of 1150 °C for different durations. The first cracking behavior was detected after exposure for 10 hr thermal treatment duration (i.e., horizontal cracks within the ceramic layers), whereas cracks at TGO/YSZ interface were developed after 100 hr treatment condition. These horizontal cracks can directly contribute to spallation and delamination of coatings out from substrate surface. Also, worthily mentioned that some vertical cracks were detected at the YSZ columnar interfaces for the 10 hr and 100 hr treatment conditions for the single layer YSZ TBC, which can influence the thermal barrier efficiency (see Figure 6 (c, d)). Dong et al. [1] investigated the TGO thickness effect on failure of plasma-sprayed TBCs and reported that the critical TGO thickness was nearly 6 µm. Hence, the first cracking behavior (10 hr) was detected at TGO thickness lower than critical value (report, ed in literature), and the crack formation can be attributed to the thermally induced residual stresses developed within TBCs during thermal loading due to thermal expansion mismatch between the TBC’ layers. For instance, the thermal expansion coefficient (CTE) of ceramic layers (about 10.4 × 10−6 K−1 and 11 × 10−6 K−1 for GZ and YSZ, respectively [2]) is typically lower than that of both substrate and bond coat (about 15 × 10-6 K−1 [3]), which leads to higher thermal stresses in the TBC system [1,4]. The second cracking behavior (cracking at TGO/YSZ interface) was detected at 100 hr thermal loading condition was driven by oxidation-induced volume increase of the TGO above critical value (approximately, 6 µm)  along with the differences in the coefficient of thermal expansion between TGO and TBC layers [1,4]. Also, it was noticed that microcracks linking up takes place at the 100 hr thermal treatment condition, which leads to final delamination of the coatings. It is known that TGO growth is one of the most important factors that lead to the failure of TBCs due to the induced stress and strain fields. The growth of TGO layer considerably changes stress intensity and its distribution [1]. For instance, compressive stresses are present at the valleys and tensile stresses at the peaks with thin TGO, while the stresses nature in the ceramic layer (YSZ) valley alters from compression to tension as TGO becomes thicker. Hence, TGO growth can significantly affect the interface structure between top and bond coat leading to reduced interfacial adhesion strength and delamination of the coatings [1].

  1. For the crystalline size: Authors do not indicate whether they have evaluated the crystalline size using (for example) Scherrer equation. I assume that instrumental broadening has been taken into account. Scherrer equation assumes that peak broadening is only due to size. However, another possible mechanism is due to incoherent strain. Authors, should make a clear point on that.

Thanks for your comment more clarification was added to the revised manuscript (XRD analysis section).

Rietveld analysis was performed by refining the crystal parameters and calculating the crystallite size to achieve good of fitness (GOF) within 5 % between the experimentally determined X-ray diffractions and the corresponding calculated ones. The crystallite size was calculated according to Lorentzian peak fitting function available in the TOPAS software. Different factors affect peak broadening including instrumental and sample contributions (e.g., source of radiation, misalignment of the diffractometer, crystallite size, micro-strain, structural faults, etc.). However, in the conducted analysis it is considered that the peak broadening coming from crystallite size evolution. Since all the samples were tested using the same XRD machine and testing parameters, the instrumental effect contribution is equal for all the samples.

  1. For the grain size: Authors provide a mean value for the grain size. However, mean value implies that the distribution of grains is normal. Apart for the case of YSZ, where the distribution may be considered as symmetrical single-mode to some extent, in all other cases, it seems that grain distribution is multi-modal. A median value is more appropriate in these cases. It would be interesting if authors could make a peak analysis, and see whether some distributions are preserved despite the thermal treatment.

Many thanks for your valuable comment. The authors would like to investigate the effect of thermal treatment on the grain growth and formation mechanisms activated as a response for the applied heat treatment. Hence, the mean value of the grain size is appropriate to represent the influence of thermal treatment on the size of the grains. Most of the data are following normal distribution specially for the bond coat and YSZ layers, which makes it more convenient to use the mean value of grain size to indicate the thermal loading effect.

  1. Minor corrections:

  1. Authors do not state the composition of the GZ/YSZ composite.

Ethanol based Suspensions consisted of an 8 wt.% YSZ powder (8YSZ) and a GZ powder, having an average particle size approximately 550 nm are used in the current study. The composite layer coating was produced using a mix of these two suspensions at a ratio of 50:50 with a solid load content of 25 wt. %.

Thanks for your comment, this has been stated in the sample preparation section.

  1. In page 3, section “thermal treatment” authors do not indicate whether the treatment was performed in air or in inert or oxidizing atmosphere.

Thanks for your comment, this has been clarified in the revised manuscript (thermal treatment section).

The thermal treatment was performed inside a Lenton furnace (AWF 12/12, Hope Valley, UK) under air atmosphere condition.

References

  1. Dong, H.; Yang, G.; Li, C.; Luo, X.; Li, C. Effect of TGO Thickness on Thermal Cyclic Lifetime and Failure Mode of Plasma‐sprayed TBC S. J. Am. Ceram. Soc. 2014, 97, 1226–1232.
  2. Gok, M.G.; Goller, G. State of the Art of Gadolinium Zirconate Based Thermal Barrier Coatings: Design, Processing and Characterization. Methods Film Synth. Coat. Proced. 2019.
  3. Vaßen, R.; Traeger, F.; Stöver, D. New Thermal Barrier Coatings Based on Pyrochlore/YSZ Double‐layer Systems. Int. J. Appl. Ceram. Technol. 2004, 1, 351–361.
  4. Vencl, A.A.; Mrdak, M.R. Thermal Cycling Behaviour of Plasma Sprayed NiCr-Al-Co-Y2O3 Bond Coat in Thermal Barrier Coating System. Therm. Sci. 2019, 23, 1789–1800.